# Immunogenic Potential of Selected Peptides from SARS-CoV-2 Proteins and Their Ability to Block S1/ACE-2 Binding

**DOI:** 10.3390/v17020165

**Published:** 2025-01-24

**Authors:** Lara Cristina da Silva Lima, Thiago Demetrius Woiski, Juliana Ferreira de Moura, Roberto Rosati, João Carlos Minozzo, Emeline Huk da Silva, Aline Castro Rodrigues Lucena, Bruno Cezar Antunes, Sérgio Caldas, Myrian Morato Duarte, Maurício Abreu Santos, Rubens Luiz Ferreira Gusso, Erickson Luiz de Moura, Ana Paula Santos Silva, Luciana Potzecki, Daniele Maria Ferreira, Elizabeth Soares Fernandes, Bonald Cavalcante de Figueiredo, Lauro Mera de Souza

**Affiliations:** 1Instituto de Pesquisa Pelé Pequeno Príncipe, Curitiba 80035-000, PR, Brazil; lara.pupo@hotmail.com (L.C.d.S.L.); thiago.woiski@pelepequenoprincipe.org.br (T.D.W.); robrosit@gmail.com (R.R.); emlinehuk@gmail.com (E.H.d.S.); danielemariaferreira@gmail.com (D.M.F.); bonaldf@yahoo.com.br (B.C.d.F.); 2Faculdades Pequeno Príncipe, Curitiba 80230-020, PR, Brazil; 3Departamento de Patologia Básica, Universidade Federal do Paraná, Curitiba 81531-980, PR, Brazil; 4Centro de Produção e Pesquisa de Imunobiológicos, Piraquara 83302-200, PR, Brazil; joao.minozzo@sesa.pr.gov.br (J.C.M.); rubensgusso@sesa.pr.gov.br (R.L.F.G.); erickson.moura@sesa.pr.gov.br (E.L.d.M.); anassilva@sesa.pr.gov.br (A.P.S.S.);; 5Diretoria de Pesquisa e Desenvolvimento, Fundação Ezequiel Dias, Belo Horizonte 30510-010, MG, Brazil; sergio.caldas@funed.mg.gov.br (S.C.); myrian.duarte@funed.mg.gov.br (M.M.D.);

**Keywords:** SARS-CoV-2, COVID-19, immunization, antibody, F(ab’)2, viral neutralizing

## Abstract

The first infection with severe acute respiratory syndrome coronavirus 2 (SARS-CoV-2), the virus that causes the coronavirus disease 2019 (COVID-19), occurred in December 2019. Within a single month, the disease reached other countries, spreading in a rapid and generalized manner worldwide to cause the COVID-19 pandemic. In Brazil, the number of COVID-19 cases surpassed 38 million. This study was conducted to produce antibodies against SARS-CoV-2 and investigate the immunogenic potential of synthetic peptides containing partial sequences of the main proteins (spike, membrane, and nucleocapsid proteins). In addition, we evaluated the ability of the antibodies to impair the interaction between the spike S1 protein and human ACE-2 protein, which is the main route of entry of the virus into host cells. By immunizing horses with synthetic peptides, we obtained hyperimmune sera with specific anti-SARS-CoV-2 antibodies, which were fragmented to release the F(ab’)2 portion that binds to the different SARS-CoV-2 proteins as a recombinant S1-protein and proteins from a viral lysate. The other F(ab’)2 samples also impaired the interaction between S1 protein and ACE-2 proteins, showing high potential to prevent viral spreading.

## 1. Introduction

Severe acute respiratory syndrome (SARS) is caused by coronaviruses (CoVs), a family of viruses that includes SARS-CoV-2, which was responsible for the coronavirus disease-19 (COVID-19) pandemic. Coronaviruses were first detected in 1937 and later described in 1965; however, SARS-CoV-2, first identified in December 2019 in China, is highly infectious and has created challenges for health professionals. The response to SARS-CoV-2 infection varies broadly, ranging from asymptomatic or mild cold-like reactions to severe and potentially deadly pneumonia [1]. Transmission occurs through the respiratory tract via secretions resulting from coughing, sneezing, and/or a runny nose [2]. Aerial transmission, along with its rapid multiplication, has resulted in a rapid spread of the virus, whose reproduction number (R0) is approximately 2.4 but can vary according to the population analyzed [3]. As a result, COVID-19 caused infections worldwide, leading the World Health Organization to declare COVID-19 an international emergency on 30 January 2020, only one month after the first recorded case.

SARS-CoV-2 is an RNA-positive virus belonging to the β-coronavirus group [4] with a 30 kb genome, which is much larger than those of other common RNA viruses. This endows the virus with genomic stability and prevents the introduction of catastrophic mutations. This genetic structure gives the virus a “self-correction” capacity via its exoribonuclease activity, ensuring replication of the viral genome and the ability to correct errors occurring during transcription [5].

The virus produces approximately 29 described proteins, the most relevant of which are the structural glycoprotein spike (S), nucleocapsid (N), membrane (M), and envelope (E). The virus has tropism to epithelial cells in the respiratory tract, with the trimeric spike protein as the main factor involved in virus invasion into host cells through its interaction with angiotensin-converting enzyme 2 (ACE-2) and subsequent entry through endosomes or fusion to the plasma membrane [3,6]. Other less important mechanisms of host cell invasion include binding of the viral E protein to porphyrins, allowing it to cross the cell membrane. This variability in cell invasion may be associated with its high infection rate [7].

The SARS-CoV-2 spike protein has two main substructures: a receptor-binding subunit (S1) and a fusion subunit (S2). The S1 fragment includes a receptor-binding domain (RBD), the binding site for the human ACE-2 protein [8]. In its pre-fusion conformation, the S1 subunit has four domains: an N-terminal domain, an RBD, and two carboxy-terminal domains. These domains surround the S2 subunit, forming a central helical bundle with the heptad repeat 1 domain of the S2 fragment projected towards the membrane [9]. The binding of S protein to ACE-2, mediated by the RBD, triggers a conformational change in the spike trimer. This alters the RBD structure, priming the S2 subunit for subsequent conformational transitions and release of the fusion machinery, ultimately leading to cell invasion [10].

Given the critical need to identify neutralizing agents against SARS-CoV-2, we explored the potential of antibodies to impair the binding of S protein to ACE-2. To achieve high yields, antibodies were generated in horses. This study, as part of a larger research effort, was performed to evaluate the ability of specific regions of key SARS-CoV-2 proteins to stimulate the immune system to produce neutralizing antibodies capable of recognizing these viral proteins and impairing their spread.

Equine sera raised against different antigens were combined to enhance efficacy to enable further therapeutic applications, including rapid detection of viral proteins. F(ab’)2 fragments from these antibodies were assayed for their specificity and ability to inhibit the interaction between S-protein and the ACE2 receptor. Our preliminary results show a potential for therapeutic applications.

## 2. Materials and Methods

### 2.1. Peptides Selection

Amino acid (aa) sequences of SARS-CoV-2 proteins were retrieved from the GISAID database (Munich, Germany) [11], and the spike (S), membrane (M), and nucleocapsid (N) proteins were selected for immunogenic peptide selection. Protein sequences from all available strains were aligned using the Clustal Omega 1.2.4 tool (Hinxton, UK) [12]. For initial peptide selection, alignments were inspected to prioritize regions with a low variation frequency. The identified regions were analyzed using the BebiPred 2.0 tool (Lyngby, Denmark) [13], and peptides of interest, containing 10–12 aa, were identified based on their high immunogenicity (Epitope Threshold so that there are 6+ positive residues in BebiPred 2.0).

In the second peptide selection step, sequences of SARS-CoV-2 proteins were evaluated using the AbDesigner algorithm (Bethesda, MD, USA) [14]. Regions of interest were evaluated based on the Ig-score, uniqueness, and conservation for selecting the peptide sequences with 10–12 aa. For peptides without a cysteine (Cys) residue in their N-terminal region, a Cys was added to allow conjugation to the carrier protein. All selected peptides were characterized for specificity against SARS virus sequences (number of residues different from the SARS homologous protein) and specificity (number of different residues vs. proteins from other organisms). Peptides were synthesized by Aminotech Research and Development (Sorocaba, Brazil).

### 2.2. Antigen Preparation

The sulfhydryl group of Cys present in the peptides was coupled to activated bovine serum albumin (BSA) (Sigma-Adrich, St. Louis, MO, USA) obtained through the cross-linker maleimide NHS ester (SMCC) (Sigma-Adrich, St. Louis, MO, USA). Activated BSA was obtained by adding 0.1 mL of SMCC dimethylformamide (Sigma-Adrich, St. Louis, MO, USA) solution (4.8 mg/mL) to 1 mL BSA in ultrapure water (Merck Milli-Q, Darmstadt, Germany (10 mg/mL), followed by mild magnetic stirring for 1 h at room temperature (RT) under protection from light. A PD10 desalting column (Cytiva, Marlborough, MA, USA) was used to separate activated and non-activated BSA. Sequential fractions were collected, and those corresponding to the absorbance peak at 280 nm (reference to BSA-SMCC) were used for subsequent experiments. For activated BSA-peptide coupling, 0.1 mL of the solution was added to an ultrapure peptide in water (10 mg/mL) under mild magnetic stirring for 30 min at RT and protected from light. Finally, the free amino groups were blocked with a Cys solution (1 mM), diluted at a protein concentration of 0.750 mg/mL, aliquoted, and stored at −80 °C (Indrel, Brazil).

### 2.3. Animal Immunization

Animal maintenance and immunization were performed at the *Centro de Pesquisa e Produção de Imunobiológicos (CPPI-Piraquara, Paraná, Brazil)*, according to institutional protocols approved by the Animal Research Ethics Committee (CEUA/SESA-CPPI n. 02/2020). The animals were immunized with different peptide preparations produced with specific sequences of proteins from SARS-CoV-2, most of which were present in the S protein and the M and N proteins. Peptides corresponding to S protein were from sites considered key for the interaction of S1/ACE-2 proteins; thus, they were injected individually and into multiple horses. Those corresponding to M and N proteins, which are not thought to contain a critical position, were inoculated together and grouped as shown in Table 1.

Seventeen healthy horses (4–5 years old), weighing 400–500 kg each, were immunized intramuscularly at 14-day intervals (Table 1). The first emulsion injection was prepared using 1 mL of antigen solution (0.750 mg/mL) and 1 mL of Freund’s complete adjuvant (Sigma-Adrich, St. Louis, MO, USA). The remaining injections were prepared using Freund’s incomplete adjuvant solution (Sigma-Adrich, St. Louis, MO, USA). The immunological response of each animal was monitored according to the plasma antibody concentration, which was monitored using enzyme-linked immunosorbent assay (ELISA) against the peptides.

First, to evaluate the immune response time, animals in group 1 (Table 1) were immunized with peptides A, B, and D–L in three separate immunization cycles. T0 blood samples were collected before the immunization cycles (used as a negative control) and after nine doses of the immunizer (C1). After a two-month rest period, the animals received a two-dose booster (C2) before sample collection; after two months, they received another three-dose booster (C3) followed by a blood draw (Figure 1). After analysis of group 1 results, group 2 animals received 14 doses of immunization continuously before blood collection (Figure 1). Sera samples were separated and stored at −20 °C (Indrel, Brasil). Animals in group 2 received peptides A, B, C, F, G, J, K, and L.

### 2.4. F(ab’)2 Fragment Purification

Before antibody enrichment and digestion, serum samples were combined with different compositions according to the immunization cycles (Table 2). To prepare F(ab’)2 from IgG, the protocol used to produce concentrated serum [IgG (Fab’2) equine] ANTI-SRCV2 on a laboratory scale was adapted from an antibody fragmentation with pepsin digestion protocol [15,16]. To 200 mL of sera mixtures, the same volume of phosphate-buffered saline (PBS) (Sigma-Adrich, St. Louis, MO, USA) was added, and the samples were subjected to enzymatic hydrolysis using 1.2 g pepsin (Sigma-Adrich, St. Louis, MO, USA) (pH 3.0–3.2) at 25 °C for 2 h under magnetic stirring in a water bath (Fisatom, Brazil). For precipitation, the pH was adjusted to 4.3, and ammonium sulfate 12% *w*/*v* (SRE0001-5G, Sigma-Aldrich, St. Louis, MO, USA) was added slowly, the temperature was increased to 55 °C, and the samples were stirred for 1 h without heating until reaching 37 °C, followed by stirring for another 30 min. The total volume was divided in half and filtered using a vacuum filter system (LC5430516, Corning^®^ Bottle-top vacuum filter system, 0.45 µm; Corning, Inc., Corning, NY, USA) connected to a vacuum pump (NB86 Laboport, KNF Group, Freiburg-Munzingen, Germany). For the second precipitate, under magnetic stirring, the pH was adjusted to 7.0–7.2, and ammonium sulfate was added to a final concentration of 32% *w*/*v* (complementing the initial amount added). After 30 min, the solution was incubated at RT for 16–18 h. The solution was resuspended under magnetic stirring, and the sample was separated (35 mL) into conical 50 mL tubes. The samples were centrifuged (NOVA Centrifugal Fixed Rotor, Kingwood, TX, USA) at 4 °C for 15 min at 1140× *g*. The precipitates were re-constituted in 10 mL of filtered PBS (0.22 µm) (Sigma-Adrich, St. Louis, MO, USA), homogenized, and filtered (Amicon^®^ Ultra-15 system, UFC903008 30 kDa; Merck, Kenilworth, NJ, USA) at 4 °C, 5000× *g* for 40 min. The initial volume was reduced from 12 to 1 mL, reconstituted in filtered PBS (0.22 µm), and stored at −20 °C. The protein concentration was quantified using a NanoDrop Microvolume Spectrophotometer (Thermo Fisher Scientific, Waltham, MA, USA).

### 2.5. Polyacrylamide Gel Electrophoresis

To verify the production of antibodies and purity of F(ab’)2 fragments, serum and purified F(ab’)2 were separated using 1D-polyacrylamide gel electrophoresis (PAGE; 15% [*v*/*v*] acrylamide). Proteins were diluted in 2X non-denaturing (125 mM Tris-HCl pH 6.8, 20% glycerol, 4% sodium dodecyl sulfate (SDS), 0.1% bromophenol blue, Sigma-Adrich, St. Louis, MO, USA) or 4X denaturing buffer (160 mM Tris-HCl pH 6.8, 10% 2-mercaptoethanol, 24% glycerol, 4% SDS, 0.1% bromophenol blue non-denaturing buffer added 350 mM dithiothreitol, Sigma-Adrich, St. Louis, MO, USA) and heated to 95 °C for 5 min, as described previously [17] with some modifications. The protein bands were stained with Coomassie Brilliant Blue R-250 (Sigma-Adrich, St. Louis, MO, USA). Protein molecular weight was estimated using a pre-stained protein ladder (Abcam, Cambridge, UK) with a molecular weight range of 5–245 kDa.

### 2.6. Western Blot Assay

The viral lysate was loaded onto a 1D 15% (*v*/*v*) polyacrylamide gel and electrophoresed. The proteins were transferred onto 0.45 µm polyvinylidene fluoride membranes (Immobilon-P, Millipore, Billerica, MA, USA) using a semidry transfer system (Power Blotter Station, Invitrogen, Carlsbad, CA, USA) set at 25 V for 30 min. The membrane was soaked in methanol and equilibrated with filter papers in transfer buffer (25 mM Tris-HCl, 192 mM glycine, and 20% methanol, Sigma-Adrich, St. Louis, MO, USA). Nonspecific binding was blocked with 5% nonfat skim milk in PBST (PBS containing 0.05% Tween 20, Sigma-Adrich, St. Louis, MO, USA) for 1 h, at RT. The membranes were incubated in the proportion 1:20 with the Fab-A, Fab-B, or Fab-C for 1 h at 25 °C, washed three times with PBST, incubated with anti-horse IgG F(ab′)2-peroxidase (SAB3700137, Sigma-Adrich, St. Louis, MO, USA) secondary antibody conjugated to horseradish peroxidase (HRP), washed three times with PBST, and incubated with o-phenylenediamine dihydrochloride. Images were obtained using an L-Pix Chemi-Imaging System (Loccus, Alagoas, Brazil).

### 2.7. Immune Response and Antigen Recognition

To measure individual immune responses (quality of IgG produced) during immunization cycles, ELISA was performed to test the animal plasma by coating 96-well plates (Thermo-Fisher Scientific, Waltham, MA, USA) with the peptides listed in Table 1 (1 µg/well); recombinants proteins S1 (Z03501-1) and human ACE-2 (Z03484-1) were purchased from GenScript (Piscataway, NJ, USA). Inactivated viral particles from SARS-CoV-2 were kindly provided by Dr. Sergio Caldas (from Fundação Ezequiel Dias) This variant was isolated in Brazil and characterized as belonging to the B.1.1.28-lineage [18]. For S1 protein recognition assays, the plates were coated with 135 ng protein per well, whereas for inactivated virus recognition assays, the plates were coated with 1 µg/well. Coating was performed overnight at −4 °C, followed by washing and blocking with 2% casein (Sigma-Adrich, St. Louis, MO, USA) in PBS for 90 min at 37 °C. Antibody samples were incubated in triplicate for 1 h at 37 °C, washed, and incubated with secondary antibodies, either anti-horse IgG–HRP or anti-horse F(ab’)2-IgG–HRP (Sigma-Adrich, St. Louis, MO, USA), for 1 h at 37 °C. After washing, the plates were incubated with 3,3’,5,5’-tetramethylbenzidine (Sigma-Adrich, St. Louis, MO, USA) for 15 min; the reaction was stopped by adding 1 M H_2_SO_4_ (Sigma-Adrich, St. Louis, MO, USA). Absorbance was measured at 450 nm using an EPOCH spectro-photometer (Agilent Technologies, Santa Clara, CA, USA).

### 2.8. ACE-2/S1 Blocking Assay

To evaluate the potential of F(ab’)2 to block the binding of S1 protein to ACE-2, a different ELISA protocol was developed using the combined F(ab’)2 from Fab-A, Fab-B, and Fab-C. Three different amounts of F(ab’)2 (1, 5, and 10 mg) were tested for each mixture. Microplates were coated with 500 ng/well of recombinant S1 protein and incubated with ACE-2 protein (60 ng/well) for 1 h at 37 °C. The test groups were first incubated with different amounts of F(ab’)2 mixtures, followed by incubation with ACE-2 protein. The protocol and buffers used in this assay were the same as those described in Section 2.6. As the secondary antibody, the anti-human IgG-Fc peroxidase antibody (00000972667; Sigma-Aldrich, St. Louis, MO, USA) was used to detect ACE-2 (which was combined with human Fc).

### 2.9. Statistical Analyses

Data normality was evaluated using the Shapiro–Wilk test. One-way analysis of variance and Dunnett’s multiple comparisons test were performed. Two-way analysis of variance and Tukey’s multiple comparisons test were used to compare the control and treated groups. Statistical analysis was performed using GraphPad Prism software for Windows, version 8.0.2 (GraphPad, Inc., La Jolla, CA, USA).

## 3. Results

### 3.1. Immune Response of Horses to Peptides

Small peptides with sequences of SARS-CoV-2 S, M, and N proteins have been used as antigens to produce antibodies against different parts of the virus, which can improve the neutralization capabilities of the yielded sera. Animals in group 1 (G1) received three cycles of immunization; after the first cycle (seven doses), heterogeneous results were observed (Figure 2A), with antigen E showing the highest titer value. After the second cycle (Table 1), a more homogeneous response was observed (Figure 2B); however, there was no substantial increase in antibody production, with titers slightly lower than those from the first cycle. Therefore, a third immunization cycle with three additional doses was performed to boost the immunogenic response. After this cycle, four antigens had higher titers (Figure 2C): peptides B and E, with sequences of S protein, including the RBD (see Table 1), and two peptides with sequences of N protein, which were used together.

Based on the immunization results for G1, animals from G2 underwent only one immunization cycle, with 14 sequential doses administered before the blood collection. Sera were obtained with much higher titers than those obtained for G1, with the antigens from S proteins A, B, and G (RBD antigen) showing higher yields, along with the combined M protein peptides J, K, and L (Figure 2D).

### 3.2. Characterization of Antibodies Produced and F(ab’)2 Enrichment

This study is part of a major effort aimed at producing hyperimmune sera with high neutralizing capabilities, enhanced by combining different antigens. Thus, the plasma samples were combined (Table 2) to increase the number of potential binding sites for IgGs on viral proteins, potentially enhancing the neutralizing efficacy. Following plasma combination, the samples were processed to enrich the IgG content; and further, the samples were digested with pepsin to generate F(ab’)2 fragments.

The production of antibodies and quality of F(ab’)2 were monitored in electro-phoresis assays. SDS-PAGE was carried out under non-reducing and reducing conditions, yielding information on entire proteins and protein fragments. The non-reducing run of total plasma proteins resulted in a typical and complex protein mixture, with bands attributed to IgGs of approximately 150 kDa (Figure 3A; lanes 1, 3, and 5). Most plasma proteins were depleted after treatment for F(ab’)2 production, observed in non-reducing lanes 2, 4, and 6 (Figure 3A), where bands of approximately 100 kDa were detected, consistent with F(ab’)2 released from the Fc fragment after pepsin cleavage.

Under reducing conditions (Figure 3B), typical IgG and F(ab’)2 bands were observed. In non-digested plasma, bands of around 60 kDa were consistent with the IgG heavy chains, and bands of approximately 25 kDa were consistent with the light chain (lanes 10, 12, and 14). F(ab’)2, previously observed at approximately 100 kDa, under reducing conditions, appeared in the gel at approximately 30 kDa, consistent with the F(ab’2) reduced fragments (Figure 3B; lanes 9, 11 and 13), which is characteristic of F(ab’)2-enriched samples.

### 3.3. Ability of F(ab’)2 to Recognize Whole Viral Proteins

The immunization cycles were followed by ELISA to determine the titer IgG production against each peptide. Despite the different titration values, all plasma samples contained antibodies against the peptide antigens.

However, to be effective, the produced antibodies should be able to bind the entire/original SARS-CoV-2 proteins. Therefore, ELISA was performed to check the ability of our F(ab’)2 preparations to recognize recombinant S1 protein and SARS-CoV-2 proteins. The results indicated that all three preparations could bind the S1 protein (Figure 4A) in a concentration-dependent manner. Small differences suggested that F(ab’)2 from the Fab-C sample was more effective in recognizing S1 protein, but no significant differences were compared with F(ab’)2 from the Fab-A and Fab-B preparations, which showed similar effectiveness.

The F(ab’)2 samples were also tested against a mixture of SARS-CoV-2 proteins obtained from a viral lysate and coated on a plate at 1 µg per well. All samples recognized all viral proteins; however, Fab-B displayed slightly inferior results (Figure 4B). As all samples gave similar results for S1 proteins, this difference observed against total viral proteins may be related to the higher ability of Fab-A and Fab-C to recognize other proteins (N and M), compared with Fab-B.

### 3.4. F(ab’)2 Sample Bound to Predicted N and M Proteins

The ability of F(ab’)2 raised against peptides to recognize the recombinant S1 protein and viral protein lysate was confirmed by ELISA assays. However, to improve their neutralizing potential, peptides from N and M proteins were also used in immunization cycles. Therefore, to evaluate the ability of F(ab’)2 samples to recognize other SARS-CoV-2 proteins, a Western blot assay was performed using a viral lysate. SDS-PAGE (Coomassie staining) revealed a typical SARS-CoV-2 protein distribution, with the main structural proteins inferred based on the molecular weight, in reducing and non-reducing conditions, as previously described [18] (Figure 5A,C). After incubation with the F(ab’)2 samples, the antibodies retained their ability to bind the predicted proteins. Notably, S protein was recognized by all F(ab’)2 samples, which was expected because most peptide antigens are derived from S protein (Figure 3B; lane 2, 4, and 6). Moreover, the Western blotting bands were consistent with the N and M proteins, confirming the ability of F(ab’)2 to recognize these proteins. S1 protein was used as a control (Figure 5B; lanes 3, 5, and 7).

Under non-reducing conditions, similar results were observed; however, the S protein produced an indistinct band, as shown in Figure 5D (lanes 2, 3, and 4). In this non-reducing analysis, despite being weakly stained, bands corresponding to the N and M proteins were also detected. These findings confirm that the antibodies generated against peptide antigens are capable of binding to the primary structural proteins of SARS-CoV-2.

### 3.5. Potential of F(ab’)2 to Block the Interaction Between S1 and ACE-2

We developed an ELISA to specifically detect the ability of experimental batches of equine F(ab’)2 to block the interaction between the human protein ACE-2 and S1 protein. As shown in Figure 6, these anti-SARS-CoV-2 F(ab’)2 preparations effectively blocked the S1/ACE-2 interaction.

The control group was incubated with S1 and ACE-2 proteins without F(ab’)2. After exposing the S1 protein to the F(ab’)2 mixtures, protein binding was significantly reduced. A concentration of 10 mg showed the greatest reduction compared to the initial value (approximately 50%), indicating a concentration-dependent blocking ability.

## 4. Discussion

SARS-CoV-2 remains an important public health issue deserving attention and continuous research, as it can be considered a model for avoiding or eliminating a new viral pandemic more quickly. Several investigations have been performed to understand viral cell infection mechanisms, demonstrating how to impair viral-cell recognition and overcome infection progress [7,8,9].

Passive immunization can help control viral spread in patients who cannot mount an effective immune response, such as patients who are immunocompromised [19]. Similarly, equine sera have been used in models of immunization against different diseases, including rabies and diphtheria, as well as in studies of antivenom therapies [20,21,22]. Therefore, equine-neutralizing antibodies may be cost-effective, scalable, and flexible tools that can complement existing COVID-19 treatments and vaccines. Their broad applicability, ease of production, and easy updates make them particularly relevant for pandemic preparedness and response, particularly in underserved regions. Indeed, F(ab’)2 has already been tested against other viral infections in animal models, confirming its ability to inhibit viral cell invasion [23]. Thus, passive immunization is a good alternative for preventing SARS-CoV-2 infection.

In this study, horses were immunized with synthetic peptides designed on conserved protein regions among SARS-CoV-2 variants, demonstrating high levels of antibodies able to react against a recombinant Spike S1 protein and SARS-CoV-2 proteins from which they were derived, as well as their ability to impair binding between the S1 protein and ACE-2 receptor, showing a potential neutralizing effect. Peptide-based vaccines have also been explored for their anti-SARS-CoV-2 potential, demonstrating their ability to stimulate long-lasting immunogenic effects in mice with high antibody production and T-cell responses [24].

In a previous study [25], horses were immunized with inactivated SARS-CoV-2 viral particles, resulting in a larger number of neutralizing antibodies, as demonstrated by the ability of F(ab’)2 to neutralize viral spread in Syrian hamsters infected with SARS-CoV-2. Additionally, the antibodies neutralized three variants of SARS-CoV-2 in vitro, suggesting the efficiency of equine antibodies in cases of COVID-19 infection where patients already have antibodies against the disease, which could enhance therapeutic effects.

Other investigations have also been conducted using inactivated viruses or entire viral proteins as immunizing agents [16,26]. We used short peptides rather than proteins or inactivated viral particles to decrease side effects, considering that anti-SARS-CoV-2 antibodies can cross-react with tissue antigens [27], as some molecular mimicry between human and viral proteins were reported and are associated with multi-organ injuries [28,29,30]. Therefore, the use of selected peptides is an attractive alternative, because it is possible to direct the production of specific antibodies, thereby reducing the risk of undesirable reactions. The sequences of the peptides used in this study did not have any homology with human proteins; thus, they are not expected to cross-react with human proteins.

Hyperimmune equine plasma has been used as an alternative to convalescent plasma; however, foreign antibodies can stimulate other adverse reactions. Additionally, antibody-dependent enhancement (ADE) requires further analysis, as it can promote undesirable infection intensification by creating other routes for host-cell invasion. Fc-receptor-dependent ADE results from the presence of cellular Fc receptors that allow the virus to bind to non-neutralizing antibodies and enter cells expressing these receptors [25]. ADE has been reported in SARS-CoV-2 infections [31,32,33,34] and is a major concern in the development of antibody-based therapies.

An alternative to overcome Fc-receptor-dependent ADE is the fragmentation of antibodies by removing the Fc region to release only the antigen-binding region, F(ab’)2. Therefore, we performed enzymatic digestion of the Fc region, using pepsin, which cleaves the Fc portion to remove small peptides from the desirable F(ab’)2. Our assays demonstrated that F(ab’)2 retained the ability to recognize all antigens, including S1-protein and proteins from the viral lysate. Additionally, these F(ab’)2 antibodies impaired the interaction of Spike S1 with human ACE-2, the main route of viral infection, which was very promising. Although this model used is limited and requires further assays to confirm viral neutralization capacity, this blocking ELISA assay offered a good option for preliminary investigation.

The S1 subunit of Spike protein is most important for viral infection because this region contains the RBD, which is fundamental for the binding of SARS-CoV-2 to the human protein ACE-2 [35,36,37]. SARS-CoV-2 shows important changes in the aa residues of its RBD region compared to those of SARS-CoV, enabling its stronger interaction with ACE-2. Thus, the RBD-ACE-2 complex is formed because of its high binding affinity [38]; disrupting this interaction may be an important approach to avoiding infections. In this regard, molecules capable of impairing this interaction are good candidates as anti-SARS-CoV-2 agents, as our synthetic peptide antigens were focused on the Spike S1 region, with peptides containing specific sequences from RBD sites (Table 1). Thus, by binding to spike-RBD, F(ab’)2 should avoid spike/ACE-2 attachment and ultimately prevent SARS-CoV-2 cell infection.

Despite the importance of RBD, this region is susceptible to mutations, supporting the importance of specific antibodies in targeting other structural proteins of the virus, such as E, M, and N, which can act synergistically with anti-Spike antibodies. Thus, we used peptides with sequences of other viral proteins. Regardless of the mutations present in the RBD, previous studies demonstrated that equine anti-SARS-CoV-2 antibodies can bind to mutated RBDs, indicating that these essential epitopes do not undergo critical changes [38].

The high specificity of monoclonal antibodies can reduce side effects; nevertheless, they can allow viral escape, as previously observed for anti-omicron antibodies [39], and are susceptible to misreactivity with antigens in case of a mutation altering the epitope. If they cross-react with human proteins, they can cause severe tissue injuries. Although our antibodies were not manufactured as monoclonal antibodies, they were highly specific because of the short aa sequences selected. To manufacture applicable hyperimmune sera, we combined sera containing different antigens to avoid viral escape, as antibodies targeting multiple sites on the virus should considerably reduce the risk of escape. To avoid antibody site competition, peptide targets from different proteins, specifically M and N proteins, were included.

Although these proteins are not related to virus and host-cell interactions, multiple targets can prevent viral invasion, as they can prevent viral access to cell receptors or hamper fusion to the host-cell membrane by steric obstruction [40]. Moreover, N protein is related to numerous complicating effects, mainly illicit acute inflammatory processes and the promotion of lung and kidney injuries [41]. Therefore, antibodies against N proteins confer additional advantages in preventing the aggravated injuries promoted by N proteins. Thus, using combined antibodies may improve the neutralizing capability, as they can act in a complementary or synergistic manner to control primary and secondary disease effects.

Indeed, the synergic effects of using peptide combinations against SARS-CoV-2 infection were previously described; however, the authors did not use peptides as antigens but explored the ability of three combined peptides to prevent the interaction between the S-protein and ACE-2 receptor, thus impairing viral infection. The peptide cocktail demonstrated excellent inhibitory and synergistic antiviral activities [42].

Therefore, immunization of horses with peptides in a controlled manner is an important tool for producing neutralizing antibodies. Horses can be re-immunized quickly with new antigens, enabling the production of new batches of equine anti-bodies tailored to emerging SARS-CoV-2 variants by engineering antigens to include conserved viral regions that can potentially neutralize multiple variants of SARS-CoV-2, including those resistant to vaccines or monoclonal antibodies.

## 5. Conclusions

Investigations and the development of neutralizing agents should be encouraged, as they expand the knowledge and arsenal of tools against COVID-19. These efforts ensure preparedness not only for future outbreaks of COVID-19 but also for related viruses that could emerge. Equine antibodies were produced through the immunization of horses with synthetic peptides and the preparation of the F(ab’)2 fragment did not affect their ability to recognize the antigens, including viral proteins. These F(ab’)2 fragments prevented S1 protein from binding to the ACE-2 receptor, as demonstrated using our developed ELISA, with the advantage of using only the proteins and not the active virus. Thus, this preliminary assay does not require special laboratory biosafety conditions. As a result, inhibition of the S1/ACE-2 interaction demonstrates the high potential of these antibodies to prevent the main route of viral host cell invasion. Further investigations should be conducted to confirm the neutralizing effects of these hyperimmune sera in animal models expressing the ACE-2 receptor. Future studies should focus on optimizing the antibody purification processes to enhance efficacy while reducing dosage requirements. Finally, these peptide sequences also demonstrate great potential for use in updating existing vaccines because they induce an immunogenic response. Therefore, a peptide-based vaccine using these sequences may enhance the effects of traditional vaccines, with fewer undesirable cross-effects.

## Figures and Tables

**Figure 1 viruses-17-00165-f001:**
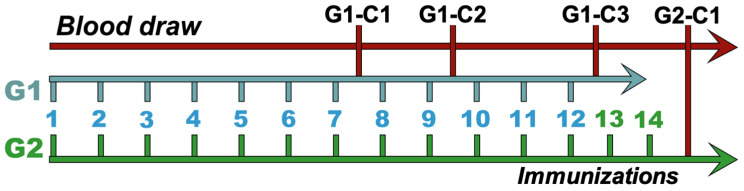
Scheme of horse immunization showing the groups (G1 and G2), number of immunizing injections (1–12 for G1 and 1–14 for G2), and time point of blood collection. G1 received three immunization cycles, with three blood draws at the end of each cycle, and G2 received a single cycle, with a blood draw at the end of the 14th antigen injection.

**Figure 2 viruses-17-00165-f002:**
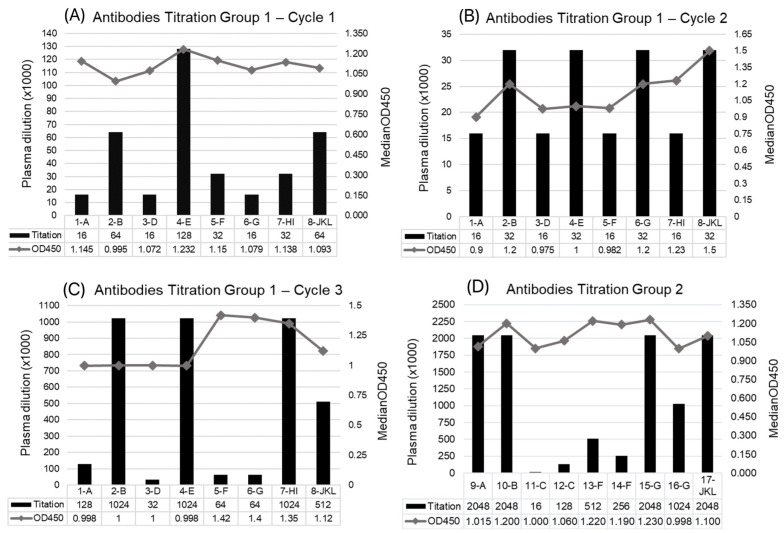
Antibody titration after each immunization cycle, showing the titer and optical density (OD) of plasma antibodies (IgG) produced for each animal from G1, after the three immunization cycles (C1, C2, and C3) and for the immunization cycle of G2 animals. The ID of each animal is presented as a sequential number (1–17) followed by a letter (A–L) indicating the peptide used for immunization, as listed in Table 1.

**Figure 3 viruses-17-00165-f003:**
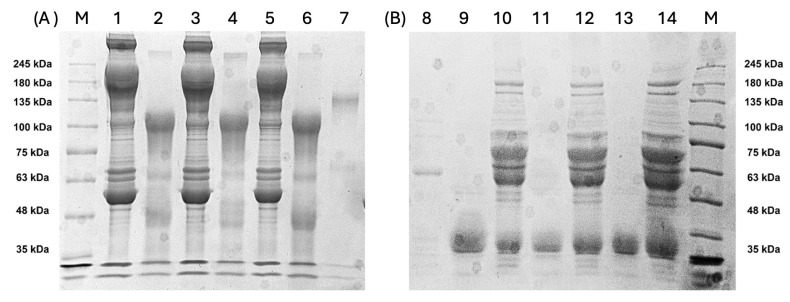
Sodium dodecyl sulfate-polyacrylamide gel electrophoresis (SDS-PAGE) analysis under (**A**) non-reducing conditions (lanes 1–8) and (**B**) reducing conditions (lanes 9–16). Lanes 1, 3, and 5 show plasma samples before F(ab’)2 preparation, with IgG detected at ~150 kDa. Lanes 2, 4, and 6 display F(ab’)2 fragments from Fab-A, Fab-B, and Fab-C, respectively, observed at ~100 kDa. Under reducing conditions, IgG fragments are evident at ~65 kDa (lanes 12, 14, and 16), while F(ab’)2 fragments appear at ~30 kDa. Lanes 7 and 8 correspond to rabbit immunoglobulin controls.

**Figure 4 viruses-17-00165-f004:**
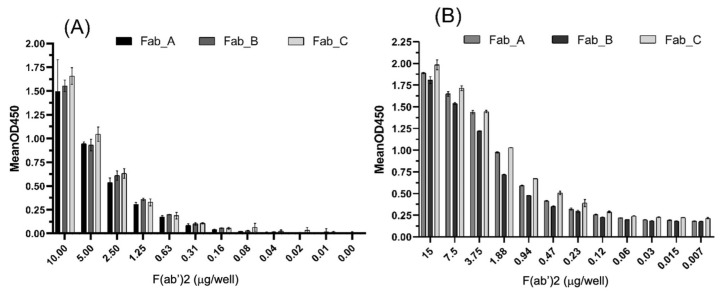
The ability of F(ab’)2 raised against peptides to recognize whole proteins from SARS-CoV-2. Total proteins were measured using colorimetric analysis and subsequently diluted in each well (μg/well). (**A**) F(ab’)2 samples were incubated against recombinant S1-protein. (**B**) Viral lysate was previously coated onto the plates, followed by incubation with F(ab’)2 samples. All results demonstrated that the F(ab’)2 samples could bind to the original proteins. Graphs were drawn using the mean (n = 3) of OD450 ± standard deviation, Two-way analysis of variance, and Tukey’s post hoc test (*p* > 0.05).

**Figure 5 viruses-17-00165-f005:**
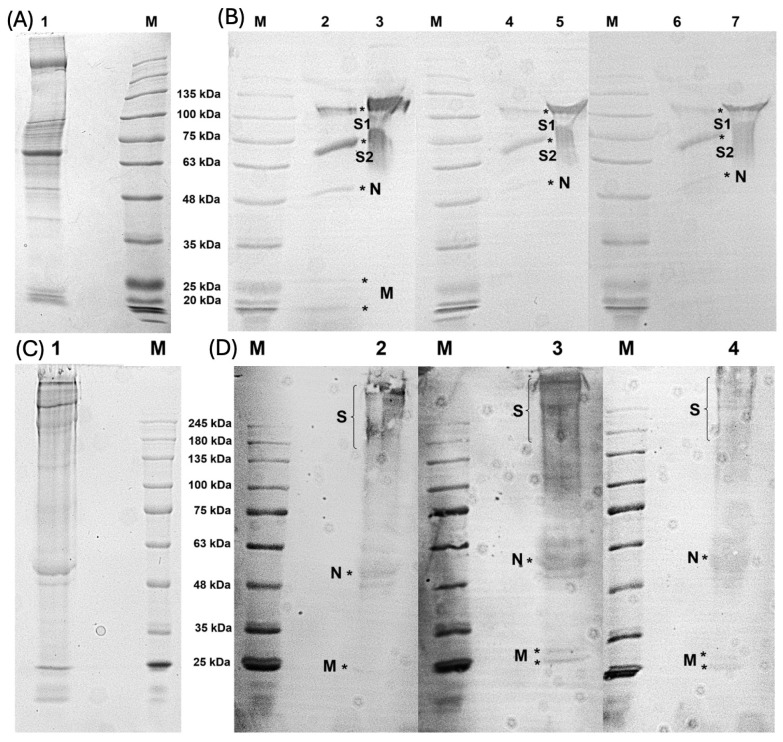
Western blot analysis of F(ab’)2 binding to SARS-CoV-2 structural proteins. (**A**) SDS-PAGE of viral lysate under reducing conditions (Lane 1). (**B**) Western blot analysis under reducing conditions showing the reactivity of F(ab’)2 fragments: Fab-A (Lanes 2–3), Fab-B (Lanes 4–5), and Fab-C (Lanes 6–7). Lanes 2, 4, and 6 correspond to viral lysate, while Lanes 3, 5, and 7 correspond to S1 protein. Estimated molecular weights: S1 ~100 kDa, S2 ~75 kDa, N ~50 kDa, and M ~20–30 kDa. (**C**) SDS-PAGE of viral lysate under non-reducing conditions (Lane 1). (**D**) Western blot analysis under non-reducing conditions showed the binding of F(ab’)2 fragments to viral proteins: Fab-A (Lane 2), Fab-B (Lane 3), and Fab-C (Lane 4). M indicates the molecular weight marker. * Indicates the position of positive bands.

**Figure 6 viruses-17-00165-f006:**
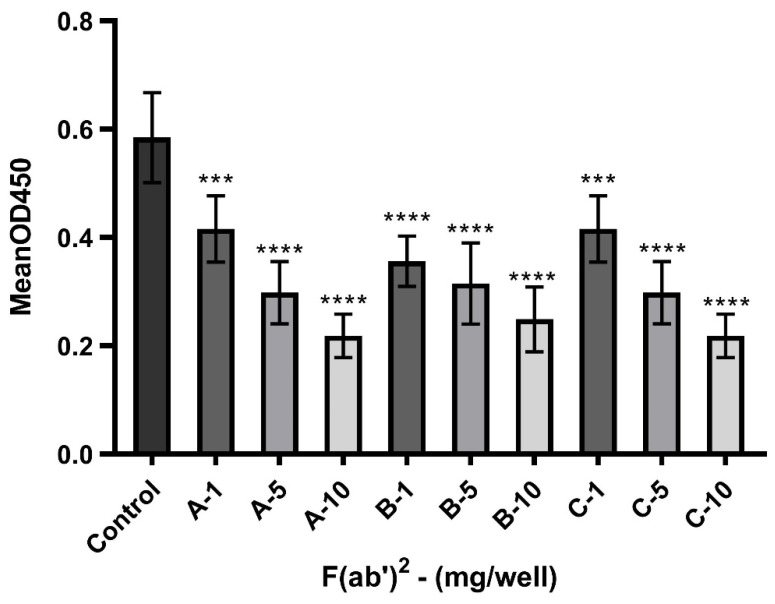
Block effect assay. Graph showing the effect of different amounts of F(ab’)2 from the Fab_A, Fab_B, and Fab_C (1, 5, and 10 mg/well) to block the interaction between Spike S1 (135 ng) protein and ACE-2 (500 ng). Bars indicating the mean ± standard deviation of OD450, one-way analysis of variance, Dunnett’s post hoc test (*** *p* = 0.0006, **** *p* < 0.00001).

**Table 1 viruses-17-00165-t001:** The animals were separated into two groups of horses (G1: 1–8; G2: 9–17) and respective peptides conjugated to the carrier protein (A–L) used for immunization. A cysteine residue was added to the N-terminal of peptides lacking this amino acid.

Immunized Animals					
Sequence Name	Peptide Sequence	Protein	AA Position	Group 1	Group 2
A	CALDPLSETKCT	S1	292–302	1-A	9-A
B	KKSTNLVKNKC	S1-RBD	528–537	2-B	10-B
C	C-LTESNKKFLP	S1	552–561	-	11-C/12-C
D	C-PTWRVYSTGS	S1	631–640	3-D	-
E	C-PDPSKPSKRS	S2	807–816	4-E	-
F	C-ADQLTPTWRVY	S1	626–636	5-F	13-F/14-F
G	C-SNNLDSKVGGN	S1-RBD	438–448	6-G	15-G/16-G
H	C-ADETQALPQRQ	N	376–386	7-HI	-
I	C-LDDKDPNFKDQ	N	339–349		
J	C-EELKKLLEQWN	M	11–21	8-JKL	
K	C-GHHLGRCDIKD	M	207–217		17-JKL
L	C-NTDHSSSSDNI	M	153–163		

**Table 2 viruses-17-00165-t002:** Plasma sample combinations.

Combination	Plasma Mix
Fab-A	G1(C1, C2, C3) + G2
Fab-B	G1(C1, C3) + G2
Fab-C	G1(C3) + G2

## Data Availability

The original contributions presented in the study are included in the article, further inquiries can be directed to the corresponding author/s.

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
