# Peer review of "Immunogenic Potential of Selected Peptides from SARS-CoV-2 Proteins and Their Ability to Block S1/ACE-2 Binding"

_viruses, 2025, doi:10.3390/v17020165_

Round 1
Reviewer 1 Report
Comments and Suggestions for Authors
In the article entitled “Immunogenic Potential of Selected Peptides from Spike Protein and Their ability to Block S1/ACE-2 interaction” by da Silva Lima, et al., the authors generated F(ab)2 from horses after immunizing with synthetic peptides derived from the Spike, Nucleocapsid, and Membrane proteins from SARS-Co-V2. These F(ab)2 bound to Spike protein and inhibited binding to ACE2 using newly-developed ELISAs. What distinguishes this paper from previous works is that they used synthetic peptides, and not whole killed virus to immunize horses. However, previous works also demonstrated the neutralizing and protective impacts of the F(ab)2 fragments in vivo. While the findings of the current study are interesting, there are a few points that should be addressed before publication:
Major Points:
11. The authors immunize with multiple peptides representing multiple targets (M, N, and S as described in Table 1). However, they only demonstrate the immune response against Spike protein. Did they generate antibodies (and subsequent F(ab)2 fragments) against the M and N proteins? Where are these data? Why were the M and N peptides included in the mix of peptides if they are inconsequential for this study? The authors state that they included these peptides so that they can be helpful in vivo if the spike protein mutates (lines 395-398), but they do not demonstrate that these anti-N and M antibodies are made. Please demonstrate whether you generated antibodies against M and N proteins and address their importance for your goals. This doesn’t require the entire virus. This could be done by Western blot or ELISA.
22. The novel ELISAs that were developed demonstrate that F(ab)2 block ACE-2 binding, but can they show that these F(ab)2 fragments block infection? This doesn’t require the original SARS-Co-V2 virus or an in vivo infection model, it can be done using a lentivirus expressing spike protein. These data would significantly increase the impact of their publication.
Author Response
Dear Reviewer,
Please find the responses for your suggestions and criticisms. We hope we could improve the quality of the manuscript accordingly.
Best regards,
Reviewer comments
In the article entitled “Immunogenic Potential of Selected Peptides from Spike Protein and Their ability to Block S1/ACE-2 interaction” by da Silva Lima, et al., the authors generated F(ab)2 from horses after immunizing with synthetic peptides derived from the Spike, Nucleocapsid, and Membrane proteins from SARS-Co-V2. These F(ab)2 bound to Spike protein and inhibited binding to ACE2 using newly-developed ELISAs. What distinguishes this paper from previous works is that they used synthetic peptides, and not whole killed virus to immunize horses. However, previous works also demonstrated the neutralizing and protective impacts of the F(ab)2 fragments in vivo. While the findings of the current study are interesting, there are a few points that should be addressed before publication:
Response – Thank you for your suggestions, we appreciated your time and knowledge employed in this review and hope we can satisfy your requested questions.
Major Points:
The authors immunize with multiple peptides representing multiple targets (M, N, and S as described in Table 1). However, they only demonstrate the immune response against Spike protein. Did they generate antibodies (and subsequent F(ab)2 fragments) against the M and N proteins? Where are these data? Why were the M and N peptides included in the mix of peptides if they are inconsequential for this study? The authors state that they included these peptides so that they can be helpful in vivo if the spike protein mutates (lines 395-398), but they do not demonstrate that these anti-N and M antibodies are made. Please demonstrate whether you generated antibodies against M and N proteins and address their importance for your goals. This doesn’t require the entire virus. This could be done by Western blot or ELISA.
Response: Alongside the E-protein, the primary structural proteins of SARS-CoV-2 include the S, M, and N proteins. Therefore, peptides corresponding to these proteins were included in our study with two main objectives. First, to enhance the potential application of these antibodies (Abs) for virus detection, thereby broadening the range of viral particles detectable in diagnostic assays. Second, according to the literature, the N-protein is highly immunogenic and associated with secondary clinical effects of SARS-CoV-2, making it a critical target for antibody-based therapies and point-of-care testing. Additionally, the M-protein, as a potentially surface-exposed protein, may contribute to neutralizing effects by increasing steric hindrance.
While antibodies against the Spike protein exhibit the greatest potential, we incorporated a higher number of peptide variants from the S protein compared to the others. This prioritization reflects the critical interaction between the ACE-2 receptor and the Spike protein, which was a central focus of our efforts.
We acknowledge the reviewers' observation regarding the lack of data on antibodies targeting proteins other than Spike. To address this, we conducted a western blot assay to demonstrate that antibodies produced in our study can also recognize other structural proteins. Please note that a viral lysate was used for this assay, as purified M- and N-proteins were not available. Nevertheless, the "in-gel" identification of these proteins was achieved by comparison with literature data. These results have been incorporated into the manuscript and can be found at lines 197–210 and 348–413, as well as in Figure 5.
The novel ELISAs that were developed demonstrate that F(ab)2 block ACE-2 binding, but can they show that these F(ab)2 fragments block infection? This doesn’t require the original SARS-Co-V2 virus or an in vivo infection model, it can be done using a lentivirus expressing spike protein. These data would significantly increase the impact of their publication.
Reponse – Blocking the interaction between ACE-2 and the Spike protein, the primary mechanism of viral entry into host cells, represents a highly promising approach to preventing infection spread. While we acknowledge that the current experiment does not conclusively demonstrate this effect, the preliminary results are highly encouraging, particularly given the simplicity of the assay and its independence from specialized biosafety laboratory conditions.
This study forms part of a broader investigation aimed at producing high quantities of F(ab’)2 fragments of equine antibodies targeting SARS-CoV-2. Unfortunately, we were unable to perform the more refined assays initially planned due to the limited availability of biosafety level 3 (BSL-3) laboratories in Brazil. We are currently awaiting the opportunity to evaluate our antibodies in vivo using ACE-2-expressing mice.
We greatly value your suggestion and will certainly consider incorporating the proposed methodology in future studies. However, at present, we lack both the expertise required for implementing this model and access to the necessary resources, such as lentiviral systems expressing the Spike protein or appropriate vectors. We appreciate your understanding and remain committed to advancing this research as opportunities become available.
Reviewer 2 Report
Comments and Suggestions for Authors
In the current manuscript, the authors produced F(ab´)2 fragments derived from antibodies (Abs) raised against different peptides selected from three structural SARS-CoV-2 proteins. They analyzed the ability of the Abs to recognize the entire protein from where the original peptide come from, inactivated viral particles, and the blockage of S1-ACE2 interaction. Although authors showed that the combination of F(ab´)2 has those abilities, major concerns arise:
Major concerns
Scientific
- Why did the authors combine all peptides together after Figure 1? Different peptides were selected but later on, there is no way to discriminate which are the best peptides, from which proteins they derived, to understand better where the activity originates.
- No neutralization ability of the Abs produced is tested. Achieving a 50% reduction in the interaction of S1-ACE2 using these Abs in an ELISA experiment doesn´t mean this would translate into neutralization. Therefore, the authors don´t prove the functionality of the F(ab´)2 fragments generated.
Format
- The goal of the project is not clearly stated.
- Before each result section, a short sentence is needed to explain the rationale of the described experiment. E.g.: Section 3.2 No explanation of how the fragments are produced, directly to the result.
- Poor description of each result section, especially in section 1.
- The discussion is too reiterative, coming after some paragraphs again to topics already discussed. Please rearrange.
General
- Line 187. What do you mean by “The SARS-CoV-2 inactivated”? Inactivated viral particles? From which variant? AgSC2-2008-2a?
- Figure 1. An immunization scheme should be added to make it clear for the reader. No explanation related to the titration.
- Figure 2. A and B are marked in the gels but not used in the text. Figure legend refers to “line” when it should to “band”.
- Figure 3. In Figure 3, the difference between FabA, B, and C is just the number of immunizations and the waiting periods but not the peptide composition. Why do authors combine all the peptides together instead of analyzing their ability independently?
- Figure 3; n=? Figure 4; n=?
- Line 312. Discuss the advantages of your strategy in comparison to other ones.
- Line 323. Not “linkage” but “interaction”.
- Line 329. Cross reacts to what?
- Line 330. Why it would aggravate the patient's condition? Other reasons to consider the peptides as a good alternative should be discussed.
- Line 340. Do you refer to ADE here? ADE is a mechanism that, in vivo, was only described for DENV infection. Do the authors have references for SARS-CoV-2 in vivo ADE?
- The fragmentation to remove the Fc region can be done to any Ab, independent of the fact of what it was raised against. What are the advantages of using synthetic peptides as immunizers instead of viral antigens or inactivated viral particles?
- Line 343. The Fc region of Ab is involved in all the effector functions apart from the neutralization itself. As the author mentioned before, conventional Abs have been used as a therapeutic in SARS-CoV-2 infected patients. If the risk of ADE is not there, it is possibly a plus to retain the Fc region. The authors didn´t discuss that.
- Line 347. Linkage is not the right concept.
- Line 350. Do the authors test the Abs obtained by the immunization with the different peptides independently on ELISA? Why is this data not shown? If available it should be disclosed since it would be the most interesting aspect of the work to know if there is a difference among peptides and peptides derived from which viral proteins are responsible for the activity.
- Line 352. Authors should show that data to claim the synergist effect.
- Line 356. Is there an overlap between the peptides described here and the one used in reference 22?
- Line 396. Which mechanisms do authors envision to explain how Ab raised against N, M, and E peptides help in neutralization? References showing neutralization of non-spike Abs?
- Line 399. Why point out the relevance of Abs being equine or any other animal? The relevant point is where those Abs bind not to be affected by the mutations acquired by the different variants. Support or delete.
- Conclusions. Next steps should be to prove neutralization in in vitro infection assays using authentic and relevant SARS-CoV-2 variants. This point is not addressed and limits the relevance of the study.
Minor concerns
- Line 137. 5´end? Do you mean N- or C-terminal?
- Table 1. Groups 1 and 2 are not described in the legend.
- Figure 2. CP and CL abbreviations are missing.
- Line 251. Cleaved by or cleaved off?
- Line 256. Section 3.4 is introduced without section 3.3
- Line 269. SARS should be written in capitals.
- Line 329. Delete “Nevertheless”.
- Line 382. Unfinished sentence.
Comments on the Quality of English Language
The paper needs to be checked thoroughly, especially regarding syntaxis.
Author Response
Dear Reviewer
Thank you for the time dedicated to reviewing our manuscript. In response to your comments and suggestions, the manuscript has been extensively revised and improved.
Comments
In the current manuscript, the authors produced F(ab´)2 fragments derived from antibodies (Abs) raised against different peptides selected from three structural SARS-CoV-2 proteins. They analyzed the ability of the Abs to recognize the entire protein from where the original peptide come from, inactivated viral particles, and the blockage of S1-ACE2 interaction. Although authors showed that the combination of F(ab´)2 has those abilities, major concerns arise:
Major concerns
Scientific
- Why did the authors combine all peptides together after Figure 1? Different peptides were selected but later on, there is no way to discriminate which are the best peptides, from which proteins they derived, to understand better where the activity originates.
Response – The initial objective of this project was to develop a scalable protocol for large-scale serum production. During preliminary immunizations, we initially attempted to use individual peptides or small groups of 2–3 peptides. However, these were later combined for practical reasons and to serve additional purposes, such as point-of-care testing. It was anticipated that antibodies (Abs) with broader targets would reduce the risk of false negatives in such applications. Furthermore, combining peptides was considered advantageous for subsequent experiments, as it would minimize the number of samples required for testing while potentially enhancing viral neutralization efficacy by reducing the likelihood of viral escape through broader epitope recognition.
Preliminary titration tests were conducted with Abs produced from individual peptides or smaller groups, but these tests were limited in scale. The blood draws for these titrations were small and performed only to monitor the production process. A portion of one of these collections, which had a reasonable titer, was used in an S1-ACE2 blocking assay. However, given that minimal differences were observed among the Abs tested, the team responsible for immunizations and sample processing opted to combine the antigens for improved efficiency.
The experiment referenced in the manuscript (and now removed) was a one-time attempt to demonstrate S1-ACE2 interaction blocking. Unfortunately, the results were inconclusive, and we did not have sufficient confidence to present or discuss them further. Additionally, we were unable to obtain new individual samples to replicate the test. We fully acknowledge that such results would have been highly valuable for discussion, and we deeply regret not having the necessary material to provide this data at present.
- No neutralization ability of the Abs produced is tested. Achieving a 50% reduction in the interaction of S1-ACE2 using these Abs in an ELISA experiment doesn´t mean this would translate into neutralization. Therefore, the authors don´t prove the functionality of the F(ab´)2 fragments generated.
Response – We appreciate the reviewer’s comment and agree with the concerns raised. However, we would like to highlight that this is a preliminary in vitro investigation, in which we successfully demonstrated the ability of antibodies (Abs) produced with small peptides to interfere with the interaction between Spike-S1 and ACE-2, as well as to recognize inactive virion particles. This study is part of a larger research effort aimed at developing neutralizing Abs on an industrial scale, which is why horses were chosen for antibody production.
Nonetheless, we encountered significant challenges in performing certain assays that are crucial for demonstrating the in vivo ability of these Abs to prevent viral spreading. We do not believe this limitation undermines the value of the current study, as it is common for many investigations to begin with in vitro assays to establish proof of concept. For instance, numerous anti-tumor studies start by testing candidate drugs in tumor cell cultures before proceeding to animal models. In the case of SARS-CoV-2, working with live viral cultures poses considerable challenges due to stringent biosafety requirements, which are not easily accessible in Brazil.
We plan to conduct further experiments to confirm the neutralizing potential of these Abs in mice expressing the ACE-2 receptor. While these results are not yet available, we conceptualized and employed a straightforward approach for preliminary evaluation, which has proven crucial in validating the potential of our Abs. This gives us confidence that they will perform effectively in future in vivo studies.
To address the reviewer’s concerns, we have revised the text to clarify any statements regarding the actual neutralization ability of the Abs at this stage of the investigation.
Format
- The goal of the project is not clearly stated.
Response – As previously mentioned, this investigation forms part of a larger research initiative aimed at producing antibodies from horses for various applications. These include serving as a potential alternative to convalescent serum, should their neutralizing ability be confirmed, as well as their use in point-of-care testing and the study of molecular interactions between the virus and host cells.
We have revised the manuscript to clarify the objectives of the project, which can now be found at lines 72–82.
- Before each result section, a short sentence is needed to explain the rationale of the described experiment. E.g.: Section 3.2 No explanation of how the fragments are produced, directly to the result.
Response – We acknowledge the reviewer’s comment and have carefully revised all result sections to address this issue appropriately.
- Poor description of each result section, especially in section 1.
Response – We have enhanced the descriptions in the results section and believe that the revisions are now sufficient for publication.
- The discussion is too reiterative, coming after some paragraphs again to topics already discussed. Please rearrange.
Response – We agree with your suggestion. The discussion section has been extensively revised to highlight our findings more concisely and to contextualize them within the existing literature. We believe these modifications have significantly improved the discussion, incorporating several important aspects from your comments, such as antibody-dependent enhancement (ADE), cross-reactivity, and the significance of the proteins included in the study. We are very pleased with these improvements and hope that you find them satisfactory.
General
- Line 187. What do you mean by “The SARS-CoV-2 inactivated”? Inactivated viral particles? From which variant? AgSC2-2008-2a?
Response – The terms “The SARS-CoV-2 inactivated,” “inactivated virus,” or “attenuated virus” have all been replaced with “inactivated viral particles” for consistency and clarity. Additionally, the text has been revised for improved clarity, and information regarding AgSC2-2008-2a (which corresponds to the B.1.1.28 lineage) has been included, along with the relevant reference from the literature.
https://doi.org/10.1016/j.molimm.2022.05.012
- Figure 1. An immunization scheme should be added to make it clear for the reader. No explanation related to the titration.
Response – A figure (1) illustrating the immunization scheme has been added, and the titration data has been presented more clearly.
- Figure 2. A and B are marked in the gels but not used in the text. Figure legend refers to “line” when it should to “band”.
Response – We have revised the text, replacing "lines" with "bands" where appropriate. Additionally, the original gels have been replaced with new ones, where the bands of interest are more clearly visible. All figures (i.e., A and B) are now properly referenced in the text.
- Figure 3. In Figure 3, the difference between FabA, B, and C is just the number of immunizations and the waiting periods but not the peptide composition. Why do authors combine all the peptides together instead of analyzing their ability independently?
Response – Fab-A, B, and C were obtained by combining peptides with some differences in immunization cycles, as demonstrated in Table 2. As mentioned previously, the decision to combine the different peptides was made with the aim of enhancing the neutralizing potential of the sera.
- Figure 3; n=? Figure 4; n=?
Response – All assays were performed in triplicate; this information has been included in the figures. Please note that Figures 3 and 4 have now been combined into a single figure, with panels designated as Figure 4A and 4B.
- Line 312. Discuss the advantages of your strategy in comparison to other ones.
Response – As previously mentioned, the discussion section has been extensively revised, and all points of criticism have been more thoroughly addressed and discussed.
- Line 323. Not “linkage” but “interaction”.
Response – “linkage” was replaced by “interactions”, accordingly.
- Line 329. Cross reacts to what?
- Line 330. Why it would aggravate the patient's condition? Other reasons to consider the peptides as a good alternative should be discussed.
Response – By "cross-reaction," we refer to the ability of antibodies produced against attenuated virions or entire viral proteins to recognize other human proteins that may share small amino acid sequences with the viral proteins. This issue is mitigated when selected peptides are used instead of whole proteins. Tissue injuries related to epitopes recognized by anti-SARS-CoV-2 antibodies support our statement. The text has been revised for clarity, and appropriate references have been added. Please refer to the updated section on lines 461-470.
- Line 340. Do you refer to ADE here? ADE is a mechanism that, in vivo, was only described for DENV infection. Do the authors have references for SARS-CoV-2 in vivo ADE?
Response – ADE has been reported in SARS-CoV-2 infections. Relevant references and an expanded discussion have been included in the text on lines 471-478
- The fragmentation to remove the Fc region can be done to any Ab, independent of the fact of what it was raised against. What are the advantages of using synthetic peptides as immunizers instead of viral antigens or inactivated viral particles?
Response – As previously mentioned, the use of synthetic peptides offers high specificity, minimizing undesirable actions. Additionally, synthetic antigens are less prone to mutations, and when mutations occur, they can be easily updated to target new variants. Furthermore, they are typically more cost-effective compared to monoclonal antibodies.
We agree with the suggestion regarding the removal of the Fc region, which can be applied to any antibodies. We have addressed this point more thoroughly in the revised text.
- Line 343. The Fc region of Ab is involved in all the effector functions apart from the neutralization itself. As the author mentioned before, conventional Abs have been used as a therapeutic in SARS-CoV-2 infected patients. If the risk of ADE is not there, it is possibly a plus to retain the Fc region. The authors didn´t discuss that.
Response – The risk of antibody-dependent enhancement (ADE) has been discussed in the literature, and we have expanded this discussion in the manuscript. Based on these concerns, we consider the fragmentation of the Fc region as the best approach to mitigate undesirable effects. This point has now been addressed in more detail, with references to relevant literature, in lines 471-485.
- Line 347. Linkage is not the right concept.
Response – Apologies for the confusion. The term "linkage" has been replaced, and we now use "interaction" or "bind" instead, with the concept being based on the receptor-binding domain (RBD).)
- Line 350. Do the authors test the Abs obtained by the immunization with the different peptides independently on ELISA? Why is this data not shown? If available it should be disclosed since it would be the most interesting aspect of the work to know if there is a difference among peptides and peptides derived from which viral proteins are responsible for the activity.
Response – I personally share the same view and agree that using individual antigens could have provided interesting results. However, as previously mentioned, the decision to combine the serum samples was made by the production and sample processing team with the goal of enhancing the neutralizing capacity. As a result, only a single ELISA assay was performed with these samples independently after the C1 blood draw. While we were in the process of designing a block-ELISA assay, the results were not convincing, and therefore, we decided not to include them in the manuscript.
- Line 352. Authors should show that data to claim the synergist effect.
Response – The suggestion of a synergistic effect was a hypothesis based on the idea that antibodies can act together, with binding to different parts of the virus potentially enhancing the ability to prevent viral entry into host cells. This is a reasonable deduction, as even if the RBD region (responsible for the Spike-ACE-2 interaction) is not blocked, binding of antibodies to different regions of the virus could create a steric hindrance effect, limiting its ability to interact with the host cell. Previous research has suggested a synergistic effect of different peptides in blocking the S1-ACE2 interaction. Therefore, we believe that presenting this idea as speculation is valid within the discussion, as it does not detract from the overall concept. While we acknowledge that further assays are required to confirm this effect, it was not the primary goal of our study. We have revised the text to soften this idea, but if necessary, we are open to removing it. Please see the updated text in lines 514-527.
- Line 356. Is there an overlap between the peptides described here and the one used in reference 22?
Response – We did not identify any overlap with the sequences from reference 22 (now ref 43).
- Line 396. Which mechanisms do authors envision to explain how Ab raised against N, M, and E peptides help in neutralization? References showing neutralization of non-spike Abs?
Response – Along with the S-protein, the main structural proteins of SARS-CoV-2 include the E-, M-, and N-proteins. Therefore, peptides derived from these proteins (except for the E-protein) were included in the study for two primary reasons: (1) to assess the potential of these antibodies (Abs) for viral detection, thereby broadening the range of viral particles that could be detected in diagnostic assays, and (2) to enhance the neutralizing capabilities of the Abs. The N-protein is known to be highly immunogenic and is associated with various secondary clinical effects of SARS-CoV-2, making it an important target for antibody therapy and antibody-based point-of-care testing. The M-protein, being a surface-exposed protein, may also play a role in assisting neutralizing effects by augmenting steric hindrance. While Abs targeting the Spike protein have the greatest potential, we used a larger number of peptide variants derived from the S-protein than from the others. Given the interaction between ACE-2 and the Spike protein, our efforts were primarily focused on Abs targeting the S-protein. Additionally, the N-protein is implicated in acute inflammatory effects, and anti-N-protein Abs could help mitigate these effects. This discussion is covered in lines 497-521, and relevant supporting references have been included.
https://doi.org/10.1038/s41577-023-00858-w
https://doi.org/10.1080/22221751.2022.2164219
- Line 399. Why point out the relevance of Abs being equine or any other animal? The relevant point is where those Abs bind not to be affected by the mutations acquired by the different variants. Support or delete.
Response – The text has been entirely reformulated to shift the focus away from equine antibodies as the primary objective.
- Next steps should be to prove neutralization in in vitro infection assays using authentic and relevant SARS-CoV-2 variants. This point is not addressed and limits the relevance of the study.
Response – Conclusions were modified accordingly.
Minor concerns
- Line 137. 5´end? Do you mean N- or C-terminal?
Response – Apologies for the confusion. The issue has been corrected, and it is now stated as the N-terminus end.
- Table 1. Groups 1 and 2 are not described in the legend.
Response – They were included
- Figure 2. CP and CL abbreviations are missing.
Response – We are sorry, this derived from Portuguese for light (“Leve”) and heavy (“Pesada”) chains. Since this figure was replaced by another better one, this was corrected.
- Line 251. Cleaved by or cleaved off?
Response – We modified this phrase for clarity.
- Line 256. Section 3.4 is introduced without section 3.3
Response – this was corrected
- Line 269. SARS should be written in capitals.
Response – This was corrected in all times it appears
- Line 329. Delete “Nevertheless”.
Response – Deleted
- Line 382. Unfinished sentence.
Response – The text has been modified for clarity.
Round 2
Reviewer 1 Report
Comments and Suggestions for Authors
For future references, PLEASE submit a clean copy of your changes. Reading this document with all of the changes, deletions, etc. made re-reviewing this manuscript extremely difficult.
Here are the specific comments:
Major:
1. The manuscript still does not contain data demonstrating that the F(ab)2 fragments block cellular infection. The authors argue that this would require BSL3. The previous review suggested using the F(ab)2 fragments to block the entry of lentiviruses overexpressing SARS-Co-V2 spike proteins, which use BLS-2. BSL requirements may differ between countries, but these data are not easily available via the internet. This is extremely disappointing, as these data would have really made this paper much more compelling.
2. Figure 4 needs and “A” and “B” and please keep the x axis consistent. My understanding is that it is the addition of F(ab)2 (ug) but on the left side, it says F(ab)2 (ug) and it says Fab (ug) on the right side. Please ensure that the label is correct.
3. Please read a clean copy of this manuscript and fix all of the grammatical errors and formatting. There are more errors in the document now than in the original document.
Author Response
For future references, PLEASE submit a clean copy of your changes. Reading this document with all of the changes, deletions, etc. made re-reviewing this manuscript extremely difficult.
Response: We sincerely apologize for this inconvenience. Many changes in the text were requested by the second reviewer, and we focused on addressing those specifically for him. However, we acknowledge that all requested alterations should have been addressed for all reviewers. This was our fault, and we deeply regret the mistake.
Here are the specific comments:
Major:
- The manuscript still does not contain data demonstrating that the F(ab)2 fragments block cellular infection. The authors argue that this would require BSL3. The previous review suggested using the F(ab)2 fragments to block the entry of lentiviruses overexpressing SARS-Co-V2 spike proteins, which use BLS-2. BSL requirements may differ between countries, but these data are not easily available via the internet. This is extremely disappointing, as these data would have really made this paper much more compelling.
Response: We deeply regret your disappointment and fully agree that incorporating this result would significantly enhance the manuscript. We had planned additional assays, such as the plaque reduction neutralization test (PRNT), which requires a BSL-3 laboratory. Unfortunately, we were unable to conduct these tests due to limitations in accessing the necessary facilities and collaborators.
Regarding lentiviruses expressing the S-protein, we currently lack the tools to perform this assay ourselves. However, we have actively sought collaborators to assist us. Given the current timing—during the vacation period for many academic and research institutions—it is unlikely we will secure the necessary support within the immediate timeframe.
Considering the three-day revision deadline provided, it will not be feasible to conduct this assay at this time. We acknowledge these limitations and sincerely apologize for the inconvenience. We hope the current version of the manuscript provides sufficient value, and we appreciate your understanding of the constraints we faced.
- Figure 4 needs and “A” and “B” and please keep the x axis consistent. My understanding is that it is the addition of F(ab)2 (ug) but on the left side, it says F(ab)2 (ug) and it says Fab (ug) on the right side. Please ensure that the label is correct.
Response: (A) and (B) were now added. The label for X-axis was changed to mg/well since it refers to this unit and the dilution rate. A text was inputted in the figure caption “Total proteins were measured using colorimetric analysis and subsequently diluted in each well (µ/well)” to make it easier to understand.
- Please read a clean copy of this manuscript and fix all of the grammatical errors and formatting. There are more errors in the document now than in the original document.
Response: We are surprised by this criticism, as the entire manuscript was professionally reviewed by a certified proofreading service (certificate attached). If the current version was perceived as worse than the first, we will need to reconsider the choice of our proofreading provider. Additionally, this version was edited by the Journal Editor, and we carefully reviewed it using Grammarly software to minimize errors.
However, due to technical limitations, we had to use a copy-and-paste approach instead of uploading the document file, which unfortunately meant that the grammatical errors were addressed, without specific changes being highlighted. We sincerely regret any inconvenience this may have caused.
